# Using Stable Sulfur Isotope to Trace Sulfur Oxidation Pathways during the Winter of 2017–2019 in Tianjin, North China

**DOI:** 10.3390/ijerph191710966

**Published:** 2022-09-02

**Authors:** Shiyuan Ding, Yingying Chen, Qinkai Li, Xiao-Dong Li

**Affiliations:** Institute of Surface-Earth System Science, School of Earth System Science, Tianjin University, Tianjin 300072, China

**Keywords:** coal replacing project, sulfur isotope, sulfate formation, oxidation pathways, Tianjin

## Abstract

After the implementation of the Coal Replacing Project (CRP) in the northern parts of China in 2017, its effect on PM_2.5_ composition is still unclear. In the study, water-soluble ionic components (WSICs) and stable sulfur isotope ratios (δ^34^S) of SO_4_^2−^ in PM_2.5_ collected during the domestic heating period before and after the implementation of CRP in Tianjin were analyzed. Results showed that the average concentrations of both PM_2.5_ and WSICs have dropped dramatically after the CRP, especially for the SO_4_^2−^ (by approximately 57–60%). After the CRP, the range of δ^34^S_sulfate_ was significantly narrowed to 4.1–7.5‰ in January 2018 and 1.4–6.1‰ in January 2019, which suggested that the sulfur source was becoming simple. It was interesting that the δ^34^S_sulfate_ value in the pollution period before the CRP was higher than that in the clean period, whereas it showed the opposite tendency after the CRP, which implied that the contribution of sea salt was high during the pollution period before the CRP. The MIXSIAR model calculated that the contributions of the transition-metal ion (TMI) oxidation and NO_2_ oxidation pathways in the three sampling stages were higher than those of the OH radical oxidation and H_2_O_2_/O_3_ oxidation pathways, indicating that the formation pathway of sulfate was mainly dominated by heterogeneous oxidation. Before the CRP, the NO_2_ oxidation pathway was the dominant sulfate oxidation pathway during a haze episode, and the TMI oxidation pathway dominated the formation of sulfates after the CRP.

## 1. Introduction

In recent years, many urban areas in China have been severely affected by haze [1,2,3]. The primary emissions, secondary formation and transformation, and stable weather conditions have been considered as the main reasons for such widespread and continuous high loading of air pollutants, including haze, over China [4,5,6]. During certain special events, the Chinese government adopted some temporary policies to pursue a great improvement in air quality, such as shutting down heavily polluting industries, prohibiting construction activities, and reducing the number of vehicles in operation [7,8]. However, the impact of these measures was limited, and the improvement in air quality could not be maintained for a long time [7,9,10]. Among many control measures taken during the “Action Plan on Prevention and Control of Air Pollution” since 2013, the Coal Replacing Project (CRP), i.e., the consumption of electricity and natural gas instead of coal for energy, was an important solution for the Chinese government to achieve the goal of reducing fine particulate matter (PM_2.5_) permanently.

It has been widely reported that the CRP has significant impacts on the energy, the economy, and the environment domains [11,12,13]. Lin and Jia (2020) concluded that the main significance of the coal-to-electricity project was not saving energy but improving air quality [12]. Ren et al. (2017) reported that the contributions of coal combustion to n-alkanes, PAHs, and OPAHs in Urumqi have been decreased from 21~75% before the implementation of CRP to 4.0~21% after the CRP [14]. Feng et al. (2020) found that the contribution of coal combustion to nitrate in the winter has been decreased by about 8% compared to the implementation of the CRP in Tianjin, and the amount of nitrate produced by the heterogeneous formation pathway has been reported to be increased after the implementation of the CRP [7]. However, most of the recent studies have focused on the role of the CRP in the emission reduction of pollutants (such as PM_2.5_, SO_2_, and NO_x_) [15,16], yet the impact of the CRP on the changes in source contributions and composition of particulate matter could not be adequately evaluated.

As a major component of PM_2.5_, the sulfate concentrations generally increase rapidly due to haze [17]. Previously, fossil fuel combustion was considered the main source of sulfate in the atmosphere [18], especially from coal burning. Although the energy structure has been changed by the CRP, fossil fuels were still the main energy composition and sulfates still played an important role. Therefore, studies on the formation mechanism of sulfate before and after the CRP could help to better evaluate the environmental effects of the CRP. Moreover, the δ^34^S sulfate could provide the oxidation pathways of SO_2_ in the atmosphere [19,20], and it has been utilized for identifying the formation pathways in haze episodes by analyzing the δ^34^S of sulfate [21,22]. Over 90% of sulfate in haze episodes was secondary sulfate and sulfate produced from SO_2_ oxidation in the atmosphere via (1) gas-phase oxidation by OH radical [23,24]; (2) aqueous oxidation by H_2_O_2_, O_3_, and transition-metal ion (TMI)-catalyzed O_2_ [25]; and (3) heterogeneous oxidation on the surface of aerosols, cloud droplets, and mineral dust by the same oxidants as aqueous oxidation [26]. Some studies also suggested that the conversion of SO_2_ to sulfate could be promoted by the coexistence of NO_x_ [17,20,25], especially during haze [24,27,28].

This study was based on the response of water-soluble ionic components (WSICs) and δ^34^S_sulfate_ in PM_2.5_ before and after the implementation of the CRP in the year from 2017 to 2019. The variation of pollution characteristics of WSICs, δ^34^S_sulfate_, and oxidation pathways of sulfate were studied to ascertain the impact of CRP on the formation of PM_2.5_ in the polluted periods. Furthermore, the improvement of air pollution due to coal-to-electricity and coal-to-natural gas policies could also be better evaluated.

## 2. Materials and Method

### 2.1. Sample Collection

PM_2.5_ sampling was conducted on the rooftop (~25 m high) of a 6-storey teaching building in Tianjin University, located at 39.11° N, 117.16° E in the center of Tianjin, which represents a typical urban environment (Figure 1) during the winter (domestic heating period) of three consecutive years, i.e., 2017, 2018, and 2019 (Appendix A). PM_2.5_ samples collection membranes were pre-combusted when collected (450 °C for 6 h). Quartz filters (23 × 18 cm) using a high-volume air sampler (TE-PM2.5HVP-BL, TISCH, Cleves, OH, USA) at a flow rate of 1.05 m^3^ min^−1^ during 8:00–18:00 (local time) for daytime and from 18:00 to next morning 8:00 for nighttime in both the seasons, and stored at −20 °C. The membranes were weighed before and after sampling using a microelectronic balance (Mettler Toledo ML 204T, Switzerland) after a 48-h equilibration in a drying vessel. One blank filter was also collected in each campaign. The samples were classified into clean (PM_2.5_ ≤ 75 μg m^−3^) and polluted (PM_2.5_ > 75 μg m^−3^) periods based on the PM_2.5_ mass concentration. Temperature (T), relative humidity (RH), and wind speed (WS) were measured at sampling points using a meteorological station (Gill MetPak, Gill Instruments, Lymington, Hampshire, UK). Gaseous pollutant (O_3_, SO_2_ and NO_2_) concentrations collected from the Tianjin Air Quality Monitoring Station data, located 1.7 km away from the sampling site.

### 2.2. Chemical Composition Analysis

Two filter cuts with a diameter of 44 mm (or four filter cuts of 26 mm in diameter if the PM_2.5_ concentration was low) were used for the analysis of WSICs. The filters were extracted into 15 mL ultrapure water (18.2 MΩ cm, Millipore, Burlington, MA, USA) under ultrasonication for 30 min at room temperature. The extractants were filtered through a 0.22 μm PTFE membrane filter and then the ions were measured using ion chromatography (ICS-5000+, Thermal, Sunnyvale, CA, USA). Cations were measured using a Dionex IonPacTM CS12A analytical column and a Dionex CRDS 600 suppressor on a conduction detector, and 20 mM MSA was used as an eluent. At a flow rate of 1.0 mL min^−1^ for anion measurement, 30 mM KOH was used as an eluent with a Dionex IonPacTM AS11-HC RFIC TM analytical column. The detection limit for WSICs was lower than 2 ng mL^−1^ with an error < 5%.

### 2.3. Determination of Sulfur Isotopic (δ^34^S) Ratios

Half of each quartz fiber filter was cut into pieces for δ^34^S analysis. They were extracted into 50 mL of ultrapure water and sonicated for 15 min, and then repeated three times. The solution was then filtered with 0.45 μm syringe filters. The filtrate was acidified to a pH < 2 by adding HCl solution and then trapped as barium sulfate (BaSO_4_) by adding droplets of supersaturated BaCl_2_ solution. The BaSO_4_ precipitation was filtered using 0.22 μm acetate filters 24 h later. After heating in an oven with step-wise heating at 200 °C for 2 h, 400 °C for 2 h, and 850 °C for 4 h, 340–450 μg BaSO_4_ were weighted and wrapped in tin capsules. The tin cups were crushed into a small size and placed into the isotope ratio mass spectrometer (MAT253, Thermo, Waltham, MA, USA). The δ^34^S of BaSO_4_ was standardized with laboratory barite standards (referenced to NBS-127 and IAEA SO-5), and reported to the international sulfur isotope standard V-CDT [22].

The δ^34^S of SO_4_^2−^ was calculated by Equation (1):δ^34^S_sulfate_ = [(^34^S/^32^S)_sample_/(^34^S/^32^S)_standard_ − 1] × 1000(1)
where the isotopic ratio of (^34^S/^32^S)_standard_ = 0.044. Overall uncertainty for δ^34^S_sulfate_ values reported here were ±0.2‰ [22].

### 2.4. MIXSIAR Model to Calculate the Formation Pathways of Sulfate Aerosols

In this estimation, isotope fractionation effects from SO_2_ to SO_4_^2−^ have to be considered. Many previous studies have shown that the fractionation factors of δ^34^S from SO_2_ to SO_4_^2−^ depend on the extent of the reaction [24,29]. Moreover, the isotope fractionation factor between SO_2_ and SO_4_^2−^ could be calculated by the Rayleigh distillation model, which was based on an instantaneous phase equilibrium process under an open system (Rayleigh 1896). In the model, the δ^34^S value of SO_2_ is a function of δ^34^S_emission_, the fraction:δ^34^S_SO2_ = δ^34^S_emission_ + ɛ_obs_ × ln(1 − SOR)(2)
where δ^34^S_SO2_ is the δ^34^S of remaining SO_2_ in the atmosphere; ɛ_obs_ is the fractionation factor between SO_2_ and SO_4_^2−^; SOR is the SO_2_ oxidation ratio; δ^34^S_emission_ is the δ^34^S of SO_2_ in emission sources, which can be estimated by the following formula based on the isotope mass balance [21]:δ^34^S_emission_ = δ^34^S_SO2_ × (1 − SOR) + δ^34^S_sulfate_ × SOR(3)
where δ^34^S_sulfate_ is the δ^34^S value of observed SO_4_^2−^ in the atmosphere. When SOR is close to 1, δ^34^S_emission_ equals to δ^34^S_sulfate_. Assuming the emitted SO_2_ is completely converted to sulfate (SOR = 1), δ^34^S_emission_ is calculated according to the linear relationship between δ^34^S_sulfate_ and SOR in each sampling stage. According to the Equations (2) and (3), the fractionation factor (ɛ_obs_) can be calculated:ɛ_obs_ = (δ^34^S_emission_ − δ^34^S_sulfate_) × SOR/[(1 − SOR) × ln(1 − SOR)](4)

Using the observed δ^34^S_sulfate_, SOR, and the estimated δ^34^S_emission_, we found the mean values of ɛ_obs_ during the sampling period were 6.6 ± 2.2‰, 0.6 ± 1.1‰, −3.1 ± 0.4‰ and 4.2 ± 1.1 ‰ during January 2017, July 2017, January 2018, and January 2019, respectively. The difference in ɛ_obs_ values reflected the different formation mechanisms of sulfate aerosols. The oxidation pathways of sulfate mainly involve gas-phase oxidation and aqueous-phase reactions [23,24,25]. Several laboratory experiments measured the fractionation factors (ɛ) for SO_2_ oxidation through H_2_O_2_ and O_3_, and these two pathways showed similar ɛ values ranging from +15.1‰ to +17.4‰ [28]. Therefore, we used ɛ_H2O2/O3_ to represent the combined isotopic effect of O_3_ and H_2_O_2_ pathways. The fractionation factors (ɛ) for SO_2_ oxidation through OH radical, H_2_O_2_/O_3_, NO_2_, and TMI pathways can be calculated [28]:ɛ_OH_ (‰) = − (0.004 ± 0.015) × T + (10.60 ± 0.73)(5)
ɛ_H2O2/O3_ (‰) = − (0.085 ± 0.004) × T + (16.51 ± 0.15)(6)
ɛ_TMI_ (‰) = − (0.237 ± 0.004) × T + (−5.039 ± 0.044)(7)
lnɛ_NO2_ (‰) = (0.2437 ± 0.0457)/T − (0.0008 ± 0.019)(8)
where T is the ambient temperature (°C). Based on the observed temperature in Tianjin during the sampling period, the fractionation factors of different oxidation pathways were obtained. Instead, the ε_obs_ should be a result of the mixing of multiple oxidation pathways [21]:ɛ_obs_ = ɛ_OH_ × f_OH_ + ɛ_H2O2/O3_ × f_H2O2/O3_ + ɛ_TMI_ × f_TMI_ + ɛ_NO2_ × f_NO2_(9)
where f_i_ is the contribution of oxidation pathway i to sulfate formation, and f_OH_ + f_H2O2/O3_ + f_TMI_ + f_NO2_ = 1. Based on Equation (8), a Bayesian isotope mixing model (ran in the Stable Isotope Analysis in the R, MIXSIAR) was used to assess the contributions of oxidation pathways to sulfate. A logical prior distribution was established in this model first, and the contribution of each oxidation pathway to the mixture was then determined.

### 2.5. HYSPLIT Backward Trajectory Analysis

This study mainly used the Hybrid Single Particle Lagrangian Integrated Trajectory Model (HYSPLIT4) of the MeteoInfo software to analyze the trajectory of backward gas and backward air mass. In this study, 100 m was regarded as the average flow field height of atmospheric boundary layer, and the backward trajectory of air mass movement is simulated every 48 h. Combined with the Pearson correlation analysis of sulfur and RH, T, SOR, tried to clarify the main factors affecting the change of S concentration in PM_2.5_.

### 2.6. Boundary Layer Height (BLH) Simulation

In order to explore the influence of BLH change on PM_2.5_ pollution in Tianjin during the pollution period, the study was conducted in the region of 116.75° to 118° east longitude and 38.5° to 39.5° north latitude, including most of Tianjin and a small part of Bohai Sea. The study area consisted of 30 grids (one grid corresponds to one height value), the resolution is 0.25° × 0.25° (about 27 km × 27 km), and the BLH per hour was obtained by connecting 30 grids (height value) together. The Data from “The Climate Data Store” (https://cds.climate.copernicus.eu/cdsapp#!/home, accessed on 5 May 2022).

## 3. Results and Discussion

### 3.1. Effect of CRP on PM_2.5_ Loading and Its Composition

As summarized in Table 1, the average concentration of PM_2.5_ in January 2018 and January 2019 dropped by approximately 60% and 57%, respectively, compared to that in January 2017. In addition, the mean value of boundary layer height (BLH) in 2017 was 313 m, which was close to 300 m in 2019 (Appendix A), indicating a great improvement in air quality after the implementation of CRP in Tianjin. In the winter of three years, three typical haze episodes (Ep1, Ep2 and Ep3) during sampling were discussed in Section 3.3.

However, the PM_2.5_ concentration two years after the implementation of CRP (January 2019) was slightly higher than that one year after the implementation of the project (January 2018), and the SO_2_ and NO_2_ concentrations in the atmosphere also showed similar annual changes (Table 1). In addition, the RH and WS were similar in January 2018 and January 2019 (Table 1). We speculated that, first, BLH in 2019 (300 m) was much smaller than that in 2018 (524 m), which makes it difficult to diffuse pollutants (Appendix A). Secondly, we assumed that in the first year of the implementation of CRP, the “coal-to-gas” and “coal-to-electricity” transformation projects were implemented strictly, and the amount of coal consumption was greatly reduced, resulting in a very low PM_2.5_ concentration in January 2018. Whereas in the second year of the CRP, some areas have the situation that the natural gas supply was not enough to support the heating demand and the consumption of coal was partly resumed to supplement the natural gas, and the coal consumption in January 2019 was slightly higher than that in January 2018 (Appendix A), which should be one of the main reasons for causing the increased PM_2.5_ loading in January 2019 compared to that in January 2018.

As shown in Figure 2, sulfate, nitrate, and ammonium (SNA) were the most important three components of WSICs, and the variation of SNA was consistent with the PM_2.5_ concentrations. The concentration of SNA in January 2017 was higher than that in January 2018 and January 2019. Otherwise, as shown in Appendix A, the proportion of sulfate in SNA decreased, and the proportion of nitrate and ammonium in SNA increased after the CRP was implemented.

### 3.2. Sulfur Isotope Composition of Sulfate Aerosols

As shown in Figure 3, before the implementation of the CRP (January-2017), the δ^34^S_sulfate_ varied widely from 3.5 to 9.5‰ with a mean value of 5.5 ± 1.6‰, and the range of δ^34^S_sulfate_ was narrowed after the CRP was implemented, which were 4.1~7.5‰ in January 2018 with a mean value of 5.6 ± 0.9‰, and 1.4~6.1‰ in January 2019 with a mean value of 4.4 ± 1.1‰. After the CRP, the range of δ^34^S_sulfate_ value gradually became smaller and the fluctuation was not high (Figure 3), which showed that with the replacement of bulk coal, the sulfur source in the atmosphere might be more stable and simpler. Moreover, the δ^34^S_sulfate_ in the pollution period (PM_2.5_ ≥ 75 μg m^−3^) before the implementation of CRP was higher than that in the clean period (PM_2.5_ < 75 μg m^−3^), while the δ^34^S_sulfate_ in the pollution period after the implementation of CRP was lower than that in the clean period. Such a decreasing trend of δ^34^S_sulfate_ from the clean period to the pollution period after the CRP might be interpreted by the changes in the contributions of various sources and formation pathways to sulfate under different atmospheric conditions.

### 3.3. Sulfur Isotopic Characteristics during the Haze Episodes

#### 3.3.1. Sulfur Isotopic Compositions of Sulfate during the Haze Episodes

To better explore the reasons for the change of δ^34^S_sulfate_ values in polluted days after the implementation of CRP, three typical haze episodes during sampling were discussed in this study. The first haze episode (Ep1) started on 1 February 2017 and lasted until 1 August 2017, with average PM_2.5_ concentrations of 217.5 μg m^−3^ and the highest value is 339.7 μg m^−3^. The average δ^34^S_sulfate_ value is 5.9‰ during Ep1, which was 25.5% higher than that in the clean period. Furthermore, the measured δ^34^S_sulfate_ displayed significant positive correlations with temperature (T) (R = 0.86; *p* < 0.01). However, it was negatively correlated with relative humidity (RH) (R = −0.45; *p* < 0.01) and SO_2_ oxidation ratio (SOR = [SO_4_^2^^−^]/[SO_4_^2^^−^ + SO_2_]) (R = −0.60; *p* < 0.01) (Figure 4). The same negative correlation between the SOR and δ^34^S_sulfate_ value during the haze events observed in Nanjing, Beijing, and other places [21,29], indicating that the δ^34^S_sulfate_ values would be affected by different SO_2_ oxidation processes. In addition, the significant negative correlation between δ^34^S_sulfate_ and RH indicated that high RH promoted the isotope fractionation process in the water phase oxidation reaction of SO_2_ [29]. Combined with the backward trajectory analysis results (Figure 5), it was found that the air mass during Ep1 mainly came from cluster 2 and cluster 3, which might be affected by the sea salt sources with high δ^34^S_sulfate_ values (21‰) (Appendix A) [30]. Therefore, before the implementation of CRP, the contribution of sea salt with high δ^34^S_sulfate_ might be one of the main reasons for the higher δ^34^S_sulfate_ in the pollution period than that in the clean period.

The haze episode 2 (Ep2) from 12 January 2018 to 14 January 2018, with average PM_2.5_ concentrations of 109.1 μg m^−3^ and the highest value was 155.5 μg m^−3^. Compared with Ep1, the mass concentration of PM_2.5_ in Ep2 decreased by 49.8%, which implies that the air quality of the Tianjin urban area has been significantly improved after the CRP. The average δ^34^S_sulfate_ value of Ep2 is 5.1‰, which was 15.7% lower than Ep1. Moreover, it was noteworthy that the δ^34^S_sulfate_ value of Ep2 was lower than that of the clean period during sampling, which was significantly different from the sampling period before the implementation of CRP (Figure 3b). In addition, the measured δ^34^S_sulfate_ of Ep2 was positively correlated with T (R = 0.70, *p* < 0.01). Since the BLH of Ep2 was significantly higher than that of Ep1 and Ep3 (Appendix A), the air mass had better diffusion ability, so the source and oxidation pathway of sulfate might change. Combined with the backward trajectory analysis (Figure 5), we found that the air mass during Ep2 mainly came from cluster 2 and cluster 4, which were mainly affected by the local sources and regional transportation in the southwest.

The haze episode 3 (Ep3) started on 10 January 2019 and lasted until 14 January 2019, with average PM_2.5_ concentrations of 127.9 μg m^−3^ and the highest value was 145.7 μg m^−3^. Compared with Ep2, the PM_2.5_ concentration of Ep3 did not decrease significantly but increased slightly, which implies that the environmental effect of the replacement of bulk coal was limited, and sulfate was still mainly derived from coal during the pollution period. Therefore, additional environmental protection policies (i.e., desulfurization and denitrification of coal) should be implemented to improve the atmospheric environment in the future. After the implementation of CRP, the air masses during Ep2 mainly came from cluster2 and cluster4, while the air masses during Ep3 mainly came from cluster1 and cluster3 (Figure 5). It was shown that the sources of air masses in the two episodes were similar, and they were mainly from local sources and regional transportation in the southwest (Figure 5). However, it was noteworthy that the δ^34^S_sulfate_ values in Ep3 decreased by 47.5% and 39.2% compared to Ep1 and Ep2, respectively. Since the regional transport of Ep2 and Ep3 is similar, it is speculated that the source has little influence on δ^34^S_sulfate_, and the variations of this value may be mainly due to isotopic fractionation during the SO_2_ conversion to SO_4_^2−^ after the CRP.

#### 3.3.2. Contributions of Oxidation Pathways to Sulfate Formation

Assuming the relative contribution of each source to sulfate aerosols was constant under the different air pollution conditions, the fractions of different sulfate formation pathways were calculated by the MIXSIAR model (Figure 6), and the contributions and uncertainties of different oxidation pathways were shown in Appendix A. It could be found that the proportion of sulfate oxidation pathway in the three stages was different. As shown in Figure 6, the contribution of the TMI oxidation pathway and NO_2_ oxidation pathway in the three sampling stages was higher than that of the OH radical oxidation pathway and H_2_O_2_/O_3_ oxidation pathway, indicating that the formation pathway of sulfate in the Tianjin urban area was mainly dominated by a heterogeneous oxidation pathway. Except for Ep2, the contribution of TMI to sulfate during sampling was consistent with other cities in China [31,32], and the OH radical pathway was also consistent with values from the previous study (19–34%) in Beijing during the polluted season (Appendix A) [32]. Thus, the source and formation pathways of sulfate have changed significantly during Ep2 when compared with the other two episodes.

Due to similar regional transport and local source contributions, we assumed that sulfate sources are similar in the winters of the 3 years. The NO_2_ oxidation pathway was the dominant sulfate oxidation pathway during Ep1 (Figure 6). Before the implementation of CRP, due to the NO_2_ pathway being more important in sulfate production under high NO_2_ and high RH conditions (Table 1) [33]. In addition, the high PM_2.5_ concentration during Ep1 (Table 1) might have amplified the formation of sulfate aerosol via the NO_2_ oxidation pathway through heterogeneous reaction, which was consistent with the research results in Beijing [17,27]. The TMIs oxidation pathway dominated the formation of sulfates during Ep2, and the contribution of Ep2 was about two times higher than that of Ep1. After the implementation of CRP, the increased contribution of the TMI pathway likely resulted from the higher efficiency of TMIs (e.g., Fe and Mn) in the atmosphere, which were emitted from local industrial sources in the haze events that enhanced the TMI oxidation rate [29].

Compared with Ep1, the contribution of sulfate formation by the OH oxidation pathway in Ep2 was significantly reduced after the CRP (Figure 6). This low contribution (5%) of gas-phase sulfate production was probably attributed to weak photochemistry in winter, high PM_2.5_ concentration, and extremely high heterogeneous and aqueous oxidations in haze episodes [21,34]. The present study highlighted that the oxidations of SO_2_ with NO_2_ and TMIs were the important formation pathways of sulfate aerosols in Tianjin, especially during the haze event.

## 4. Conclusions

After the implementation of CRP, the average concentration of PM_2.5_ in January 2018 and January 2019 dropped by approximately 60%. The trend of WSICs was consistent with PM_2.5_. The δ^34^S_sulfate_ in the clean period was lower than that in the pollution period before the CRP, which was mainly affected by sea salt sources with high δ^34^S. However, the δ^34^S_sulfate_ in the clean period was higher than that in the pollution period after the CRP, which was mainly due to the increase of heterogeneous oxidation contribution during the pollution period. The contribution of the TMI oxidation pathway and NO_2_ oxidation pathway in the three sampling stages is higher than that of the OH radical oxidation pathway and H_2_O_2_/O_3_ oxidation pathway, which shows that the formation pathway of sulfate in the Tianjin urban area is mainly dominated by heterogeneous oxidation pathway. Before the implementation of CRP, the NO_2_ oxidation pathway was the dominant sulfate oxidation pathway during Ep1, and the TMI oxidation pathway dominated the formation of sulfates after the implementation of CRP. Therefore, it should be noted that multiple energy-saving and emission reduction policies should be employed to further optimize the energy structure and therefore maximize the environmental benefits of energy conservation and emission reduction policies.

## Figures and Tables

**Figure 1 ijerph-19-10966-f001:**
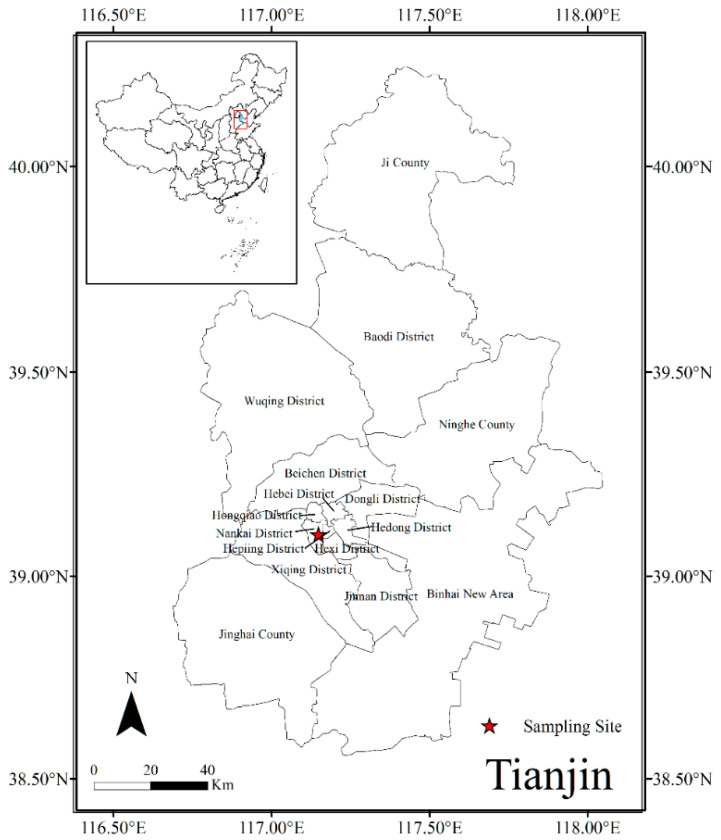
Sampling site located in the Tianjin.

**Figure 2 ijerph-19-10966-f002:**
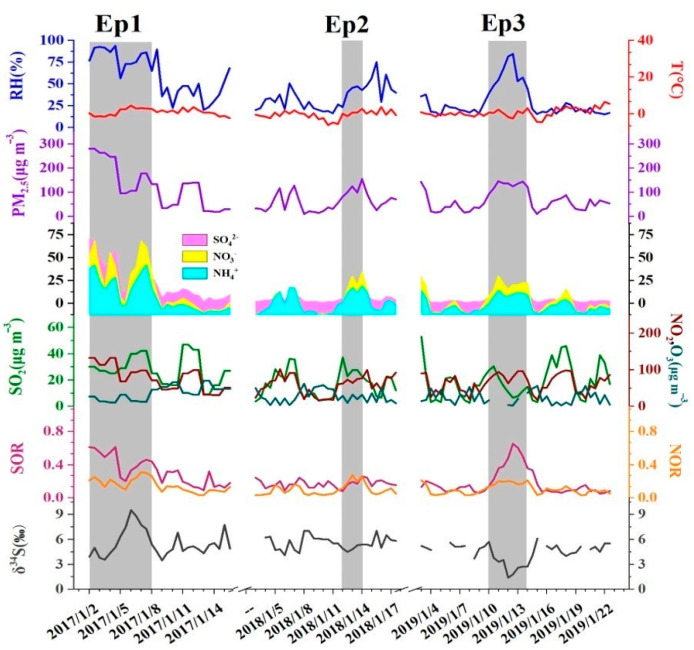
Time series of pollutant concentrations (PM_2.5_, SNA, SO_2_, NO_2_ and O_3_), SOR, NOR, δ^34^S of sulfates and the meteorological data (RH and T) during six campaigns. The gray shaded area indicated a polluted period (PM_2.5_ ≥ 75 μg m^−3^).

**Figure 3 ijerph-19-10966-f003:**
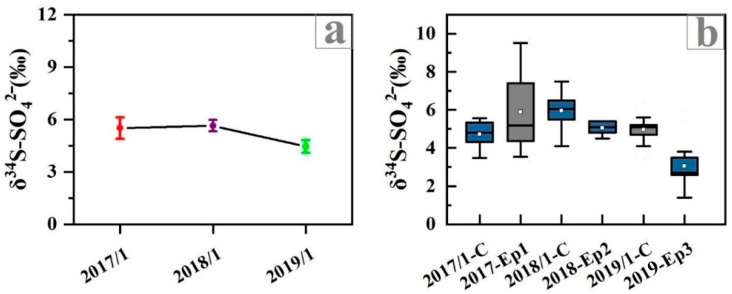
(**a**) Average δ^34^S-SO_4_^2−^during four sampling periods, (**b**) Box plots of δ^34^S-SO_4_^2−^ during the clean period (blue boxes) and polluted period (grey boxes).

**Figure 4 ijerph-19-10966-f004:**
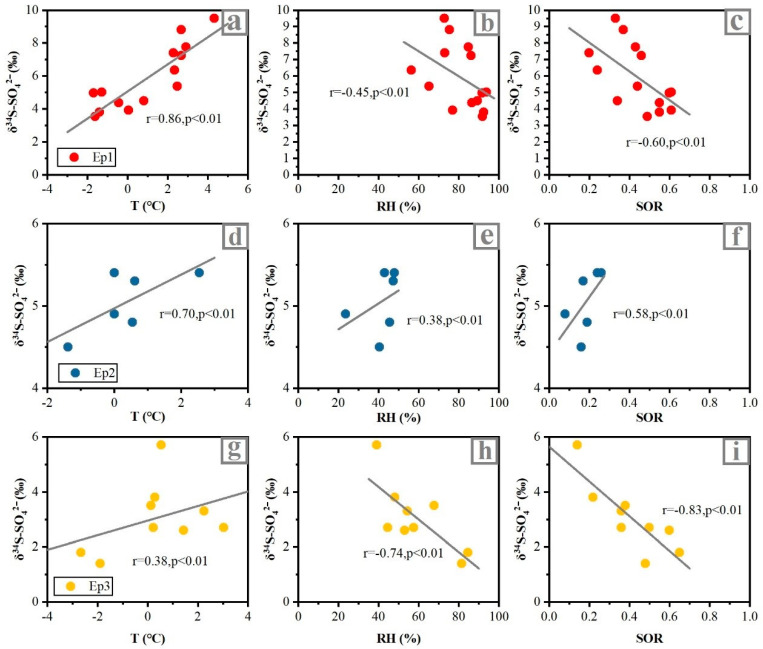
Relationships between δ^34^S-SO_4_^2−^ and temperature (T), relative humidity (RH), and SOR in Tianjin during Ep1 (**a**–**c**), Ep2 (**d**–**f**), Ep3 (**g**–**i**).

**Figure 5 ijerph-19-10966-f005:**
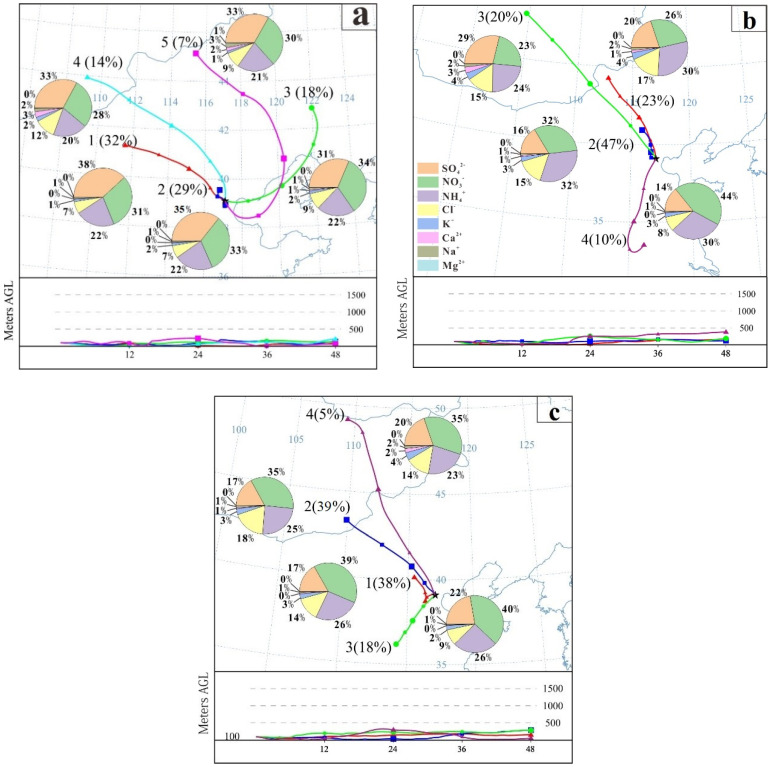
48-h backward trajectory of (**a**): January2017, (**b**): January2018, (**c**): January2019.

**Figure 6 ijerph-19-10966-f006:**
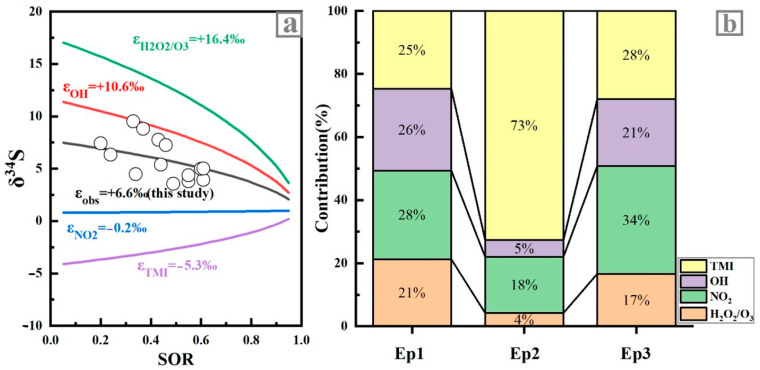
(**a**) Rayleigh distillation model of sulfate production. Black circles are the measured δ^34^S_sulfate_ values from this study. The green line indicates the δ^34^S_sulfate_ when SO_2_ is oxidized solely by H_2_O_2_ and O_3_. The red line indicates the δ^34^S_sulfate_ when SO_2_ is oxidized solely by OH. The blue line indicates the δ^34^S_sulfate_ when SO_2_ is oxidized solely by NO_2_. The purple line indicates the δ^34^S_sulfate_ when SO_2_ is oxidized solely by TMI oxidation. The black line is the best fit of observed δ^34^S_sulfate_, showing an average ε_obs_ value of +6.6 ± 2.1‰. (**b**) Contribution of each formation pathway based on the Bayesian model calculation.

**Table 1 ijerph-19-10966-t001:** PM_2.5_ (μg m^−3^), water-soluble ions concentrations (μg m^−3^), Meteorological parameters: relative humidity (RH, %), temperature (T, °C), wind speed (WS, m s^−1^), gaseous contaminants concentrations (μg m^−3^): SO_2_, NO_2_, O_3_, sulfur oxidation ratio (SOR) and nitrogen oxidation ratio (NOR) of the sampling campaigns (C: clean period; Ep: haze episode).

	January 2017			January 2018			January 2019		
	All (*n* = 28)	C (*n* = 6)	Ep1 (*n* = 14)	All (*n* = 30)	C (*n* = 19)	Ep2 (*n* = 6)	All (*n* = 40)	C (*n* = 27)	Ep3 (*n* = 9)
PM_2.5_	151.6 ± 86.2	59.7 ± 19.2	217.5 ± 72.5	60.7 ± 39.2	35.5 ± 17.8	109.1 ± 26.5	65.4 ± 43.3	39.0 ± 18.2	127.9 ± 16.7
Na^+^	0.7 ± 0.5	0.3 ± 0.1	1.1 ± 0.3	0.3 ± 0.2	0.2 ± 0.1	0.4 ± 0.1	0.3 ± 0.1	0.2 ± 0.1	0.4 ± 0.1
NH_4_^+^	15.1 ± 13.2	3.1 ± 1.6	25.3 ± 11.3	9.2 ± 7.8	4.1 ± 3.4	18.1 ± 4.8	7.6 ± 6.1	3.9 ± 2.5	15.8 ± 3.1
K^+^	1.0 ± 0.8	0.3 ± 0.1	1.7 ± 0.7	0.9 ± 0.6	0.5 ± 0.3	1.6 ± 0.3	0.9 ± 0.6	0.6 ± 0.4	1.4 ± 0.3
Ca^2+^	0.3 ± 0.3	0.4 ± 0.2	0.4 ± 0.4	0.3 ± 0.2	0.2 ± 0.2	0.3 ± 0	0.1 ± 0.1	0.1 ± 0.1	0.2 ± 0.1
Mg^2+^	0.04 ± 0.03	0.04 ± 0.04	0.04 ± 0.02	0.01 ± 0.02	0.01 ± 0.02	0.02 ± 0.00	0.03 ± 0.01	0.02 ± 0.10	0.03 ± 0.01
Cl^−^	5.2 ± 5.0	1.5 ± 0.7	8.4 ± 5.2	4.2 ± 4.3	2.0 ± 1.8	6.0 ± 1.4	4.0 ± 2.6	3.1 ± 2.3	5.0 ± 1.4
NO_3_^−^	22.3 ± 19.4	4.54 ± 2.65	37.3 ± 16.9	10.0 ± 9.8	3.82 ± 3.07	24.0 ± 8.9	11.5 ± 9.5	5.65 ± 3.83	24.1 ± 4.6
SO_4_^2−^	24.6 ± 21.2	5.3 ± 3.2	40.8 ± 18.7	5.0 ± 3.2	3.0 ± 1.7	8.4 ± 2.4	5.6 ± 5.7	2.2 ± 1.1	14.7 ± 3.7
RH	60.6 ± 24.2	28.5 ± 7.4	81.1 ± 11.4	34.8 ± 14.4	31.2 ± 16.1	41.3 ± 9.1	29.4 ± 18.3	19.4 ± 3.6	58.9 ± 15.9
T	0.9 ± 1.9	0.5 ± 1.3	1.0 ± 2.0	−0.7 ± 2.3	1.4 ± 2.5	0.4 ± 1.3	0.6 ± 2.4	0.5 ± 2.7	0.4 ± 1.8
WS	1.4 ± 0.7	2.3 ± 0.8	1.0 ± 0.4	1.5 ± 0.4	1.6 ± 0.5	1.4 ± 0.3	1.5 ± 0.7	1.7 ± 0.8	1.1 ± 0.3
SO_2_	28.1 ± 11.3	13.8 ± 3.8	31.1 ± 6.7	17.0 ± 10.1	11.0 ± 6.1	26.2 ± 6.3	18.7 ± 13.1	16.2 ± 11.3	16.2 ± 8.5
NO_2_	78.9 ± 33.8	36.5 ± 8.6	101.0 ± 25.2	57.8 ± 26.1	45.1 ± 23.1	74.7 ± 13.5	63.6 ± 24.9	52.9 ± 22.6	82.5 ± 11.8
O_3_	36.1 ± 19.5	58.8 ± 11.1	21.9 ± 12.9	27.7 ± 17.8	33.3 ± 18.8	22.0 ± 10.3	22.9 ± 16.8	26.4 ± 17.4	14.8 ± 12.9
SOR	0.3 ± 0.2	0.2 ± 0.1	0.4 ± 0.1	0.2 ± 0.1	0.2 ± 0.1	0.2 ± 0.1	0.2 ± 0.2	0.1 ± 0.1	0.4 ± 0.2
NOR	0.1 ± 0.1	0.1 ± 0.1	0.2 ± 0.1	0.1 ± 0.1	0.05 ± 0.1	0.2 ± 0.1	0.1 ± 0.1	0.1 ± 0.1	0.2 ± 0.01

## Data Availability

Data is contained within the article or Appendix A.

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
