# Peer review of "Using Stable Sulfur Isotope to Trace Sulfur Oxidation Pathways during the Winter of 2017–2019 in Tianjin, North China"

_ijerph, 2022, doi:10.3390/ijerph191710966_

Round 1

Reviewer 1 Report

Line 74 and 75. The sentence lacks clarity. The authors should think about revising it.

Line 194. Concentrations of meteorological parameters is not best expression. Revision is neede.

Author Response

Point 1: Line 74 and 75. The sentence lacks clarity. The authors should think about revising it.

Response 1: Thanks for the reviewer’s comment. We have deleted the sentence in the revised manuscript.

Point 2: Line 194. Concentrations of meteorological parameters is not best expression. Revision is neede.

Response 2: Thanks for the reviewer’s comment. We have revised the sentence. (Lines 206-207)

Lines 206-207: Figure 2. Time series of pollutants concentrations (PM2.5, SNA, SO2, NO2 and O3), SOR, NOR, δ34S of sulfates and the meteorological data (RH and T) during six campaigns.

Reviewer 2 Report

The manuscript presents the analysis of sulfur oxidation pathways based on the stable sulfur isotope ratios. There were three episodes which correspond to periods before and after the implementation of the Coal Replacing Project (CRP) in China. This is the original paper which brings novelty and could be interesting to readers.

However, there are a few problematic moments which need consideration.  

Major comments:

1. My main concern comes from the meteorological aspects of the study. As authors are dealing with rather short temporal periods (three cases less than a month each one) - the role of meteorological conditions increases. Yes, authors considered wind speed, relative humidity, air temperature and air masses movement. However, one of the biggest impacts on elevated pollution levels are provided by air temperature vertical distribution (i.e. inversions) and vertical motions. In a short period of time (less than a month), vertical structure of the atmosphere very often has the biggest influence on concentrations of pollutants in the boundary layer. That is why most results and authors' explanations in Section 3.1 and 3.2 are arguable. It is not correct to compare concentrations without considering boundary layer height (BLH) or, at least, air temperature vertical distribution and frequency of inversions. These parameters also significantly define the ability of the atmosphere to accumulate or disperse air pollution. Lower BLH causes higher concentrations. I highly recommend authors to consider BLH before trying to explain Section 3.1 and 3.2. (BLH data could be found in reanalysis from ECMWF; or be modelled; or estimated using radio-soundings). Reading further, my concerns increased after I looked at "Meters AGL" in fig. 5. It is possible that higher concentrations in 2017, and the difference between 2018 and 2019, might be partially (or significantly) caused by lower BLH.  

2. I have some doubts about the results in Section 3.3.1 where authors deal with correlation and further explanations. Dependencies from Episode 2 differ from Ep.1 and Ep.3. Despite p-values being shown, the length of the time series is too small for using Pearson correlation to Ep.2 (I decided that it was Pearson because there is no clarification in the text). For Ep.2 Pearson correlation is unreliable just because the number of points is too small. It might be the reason for the different dependency from RH and  SOR. Authors need to think about how to discuss this.  

3. It was used the method for building backward trajectories, however there are no links and explanations about the tool. It seems that it was the HYSPLIT model, but it is only my guess. Authors must describe the tool which they used for building backward trajectories including input parameters or crucial options they selected. Was it web-based or model setup? It is better to add this information to section Materials and Method as separate section 2.5 together with the details on correlation (was it Pearson or not), what means "significant", and other statistical details. And, of course, with all required links/ references to the described tool.  

Minor comments:

1. It is better to give the description of episodes at the very beginning of Results. When I came to Table 1, I did not understand what is "Ep1", "Ep2", "Ep3". The description I found only in section 3.3  

2. Also, I still do not understand what "C" means in the header of Table 1.  

3. Line 93. What is the reason for taking the threshold of 75 μg m-3 for PM2.5? Is it the value from legislation? If not, why have you decided to use this classification?    

4. Could you please give the explanation of what WSIC is when it is first mentioned in the main body of the manuscript (not only in Abstract).

Round 2

Reviewer 2 Report

No more comments